
# A comprehensive estimate for loss of atmospheric carbon tetrachloride (CCl4) to the ocean

J. H. Butler[1], S. A. Yvon-Lewis[2,7], J. M. Lobert[3,7], D. B. King[4,7], S. A. Montzka[1], J. L. Bullister[5], V. Koropalov[6], J. W. Elkins[1], B. D. Hall[1], L. Hu[1,2] and Y. Liu[2,8]

[1]Global Monitoring Division, NOAA Earth System Research Laboratory, Boulder, 80305, USA
[2]Department of Oceanography, Texas A&M University, College Station, 77840, USA
[3]Entegris Inc., Franklin, 02038, USA
[4]Chemistry Department, Drexel University, Philadelphia, 19104, USA
[5]NOAA Pacific Marine and Environmental Laboratory, Seattle, 98115, USA
[6]Roshydromet, Moscow, 123242, Russia
[7]Cooperative Institute for Research in Environmental Sciences, University of Colorado, Boulder, 80309, USA
[8]Marine Chemistry & Geochemistry, Woods Hole Oceanographic Institution, Massachusetts

*Correspondence to*: James H. Butler (james.h.butler@noaa.gov)

**Abstract.** Extensive undersaturations of carbon tetrachloride (CCl4) in Pacific, Atlantic, and Southern Ocean surface waters indicate that atmospheric CCl4 is consumed in large amounts by the ocean. Observations made on 16 research cruises between 1987 and 2010, ranging in latitude from 60° N to 77° S, show that negative saturations extend over most of the surface ocean. Corrected for physical effects associated with radiative heat flux, mixing, and air injection, these anomalies were commonly of the order of -5% to -10%, with no clear relationship with temperature, productivity, or other gross surface water characteristics other than being more negative in association with upwelling. The atmospheric flux required to sustain these undersaturations is 11 (7–14) Gg y$^{-1}$, a loss rate implying a partial atmospheric lifetime with respect to the oceanic loss of 209 (157–313) y and that ~16 (10–21) % of atmospheric CCl4 is lost to the ocean. Although CCl4 hydrolyses in seawater, published hydrolysis rates for this gas are too slow to support such large undersaturations, given our current understanding of air–sea gas exchange rates. The even larger undersaturations in intermediate depth waters associated with reduced oxygen levels, observed in this study and by other investigators, strongly suggest that CCl4 is ubiquitously consumed at mid-depth, presumably by microbiota. Although this subsurface sink creates a gradient that drives a downward flux of CCl4, the gradient alone is not sufficient to explain the observed surface undersaturations. Since known chemical losses are likewise insufficient to sustain the observed undersaturations, this suggests a possible biological sink for CCl4 also in surface or near-surface waters of the ocean.

## 1 Introduction

CCl4 is a strong ozone-depleting gas for which production for dispersive use (e.g., fire suppression, dry cleaning, fumigation) has been banned through the Montreal Protocol (1987) and its amendments and adjustments. Although the concentration of



atmospheric $CCl_4$ has been declining in the atmosphere since the early 1990s, its rate of decline is slower than predicted from estimates of emissions suggested by production data reported to the Ozone Secretariat (<11 Gg/yr since 2007; Carpenter and Reimann et al., 2014) and its atmospheric lifetime (e.g, Liang et al 2014, Carpenter and Reimann 2014). The dominant loss for $CCl_4$ is through photolysis in the upper atmosphere, which, based on the most recent evaluations, would yield an

atmospheric lifetime of 44 (36–58) y (Laube et al. 2013, Carlon et al. 2010, Volk et al. 1997). The oceanic sink, previously calculated as 94 (82–191) y (Yvon-Lewis and Butler 2002), also removes significant amounts of $CCl_4$ from the atmosphere. The $CCl_4$ lifetime owing to uptake by soils, previously determined at 90–195 y, but recently assessed at 375 (288–536) y, is considered a lesser and more uncertain component (Happell et al. 2014, Rhew and Happell 2016). These additional sinks had previously brought the overall calculated lifetime of $CCl_4$ in the atmosphere down to 23–35y. The oceanic sink for $CCl_4$

determined by Yvon-Lewis and Butler (2002) and used in subsequent Scientific Assessments of Ozone Depletion (e.g., Carpenter and Reimann 2014), however, was based almost entirely on surface data from four research cruises in the Pacific Ocean from 1987–1992 (Butler et al, 1997). Considerable data exist for a deficit of $CCl_4$ in deeper ocean waters, particularly those associated with low oxygen (e.g. Krysell and Wallace 1994, Tanhua and Olsson 2005). This study focuses on surface data from the original four cruises and a dozen additional expeditions to enhance the earlier analysis, to examine the oceanic

sink for potential sampling and analytical biases, to evaluate the potential cause of the sink, and to provide more confidence in the estimated mean rate of atmospheric $CCl_4$ removal by the ocean. This study also takes advantage of significant improvements in determining air–sea exchange rates, which have a substantial impact on the final estimate. Finally, we draw on hydrographic data from selected cruises to underscore the role of subsurface processes. With extensive surface data from numerous cruises coursing three of the world's major oceans, we provide here a more representative picture of oceanic

removal of this gas from the atmosphere.

## 2 Sampling and Analysis

$CCl_4$ mole fractions in air were measured hourly in equilibrated surface water and the atmosphere on most of 16 research cruises, crossing many of the major ocean basins over a period of 23 years (1987–2010; Fig. 1, Table 1). (On four cruises, SAGA II in the West Pacific, CLIVAR-01 in the Southern Ocean, and A16N and A16S in the Atlantic, we resorted to

sampling daily from surface Niskin bottles; WOCE P18 data, likewise from Niskin bottles, were used only for depth profile analyses and were not accompanied by continuous air measurements.) On the remaining cruises, air samples were collected from the ship's bow and surface samples were obtained with an underway, Weiss-type equilibrator (e.g., Johnson et al. 1997, Butler et al. 1988). Gases in all samples were separated by gas chromatography on OV-101 or similar columns, some packed, most capillary, and detected with electron capture detectors (GC-ECD) or mass spectrometers (GCMS; Hewlett

Packard or Agilent 5971 or 5973). Both types of detectors and different columns were used on some cruises to evaluate potential analytical biases. To evaluate potential bias introduced by the equilibrator, surface samples from hydrocasts or bucket samples were obtained and analyzed on several occasions. Gases were extracted from these samples with a purge-



and-trap technique (Bullister and Weiss 1988, Yvon-Lewis et al 2003), and subsequently analyzed by GCMS or GC-ECD. Results from these grab samples agreed well with those from underway equilibrated surface water (Fig. 2).

Full-depth water column profiles of dissolved $CCl_4$ concentrations, along with CFC-11 and CFC-12, were also collected on several expeditions during this period. These profiles typically consisted of measurements made on discrete water samples collected at 24–36 depths, using purge-and-trap techniques and analyzed with GC-ECD. Depth profiles of $CCl_4$ were obtained in some instances to identify potential zones of $CCl_4$ loss relative to that of CFC-11, a gas of similar physical properties, but unreactive in seawater except under virtually anoxic conditions (Bullister and Lee, 1995; Shapiro et al., 1997). However, to further understand the potential cause of the surface deficits of $CCl_4$, we also evaluated extensive data from World Ocean Circulation Experiment (WOCE) Repeat Section P-18 (2008), which runs from 21ºN to 70ºS in the East Pacific Ocean, and used CFC-12 as the conservative tracer. This section runs through an extensive, well developed oxygen minimum, thus allowing a close evaluation of the correlation between dissolved $CCl_4$ and $O_2$.

## 3 Computations

"Instantaneous" $CCl_4$ fluxes were estimated from the observed difference between partial pressures of $CCl_4$ in the atmosphere and those in the surface ocean, the air–sea exchange velocity, and the solubility and diffusivity of the gas. Influences of in situ physical effects, such as warming, cooling, and mixing (Kester 1975), on $CCl_4$ saturation anomalies were minimized by subtracting saturation anomalies of CFC-11 (e.g., Butler et al. 1991). The saturation anomaly ($\Delta_g$) for a dissolved gas is expressed as the percent departure of the observed dissolved amount from equilibrium. This is computed from the difference in partial pressures:

$$\Delta_g = 100 \left( \frac{p_{gw} - p_{ga}}{p_{ga}} \right) [\%] ,\qquad(1)$$

where $p_{gw}$ and $p_{ga}$ are the partial pressures of the gas in water and air. If the saturation anomaly of a gas is positive, it indicates that the water is supersaturated and the net flux is from the ocean to the atmosphere. If it is negative, then the net flux of the gas is from the atmosphere to the ocean. However, the saturation anomaly alone is not sufficient for detecting or estimating in situ consumption of the gas in the water. This is because of physical processes that can give rise to positive or negative saturation anomalies (Kester 1975). The magnitude of this effect depends upon the diffusivity of the gas, its solubility, and the temperature dependence of its solubility. In practice, the difference in saturation anomaly for similar compounds is small; saturation anomalies for CFC-11 and CFC-12 typically differ by 1–3%. Their molecular diffusivities are not much different, nor is the temperature dependence of their solubilities, but their absolute solubility differs by a factor of 3.6 (Table 2).

Consequently, to determine if there is any consumption of a gas such as $CCl_4$ in the water, we compute its *corrected* saturation anomaly to capture that portion of the saturation anomaly that is largely free of physical influences. This is done by subtracting the CFC-11 saturation anomaly from the observed $CCl_4$ saturation anomaly. CFC-11 is chosen over CFC-12,


because its physical properties more closely resemble those of $CCl_4$ than do those of CFC-12, which is much less soluble and has a smaller change in solubility with temperature. Calculated this way, a corrected saturation anomaly that is negative indicates that the gas is probably being consumed in the water, regardless of its non-corrected anomaly. However, because of the differences in physical properties of various gases, in situ consumption is more probable if the corrected saturation

anomaly is less than -2%.

The corrected saturation anomaly of $CCl_4$ should be roughly proportional to its in situ loss or production. If we assume steady-state conditions, the loss or production rate can be calculated from the flux across the surface of the water that is required to maintain the corrected saturation anomaly:

$$F_{CCl_{4,parcel}} = \frac{K_w p_{CCl_{4,a}} A}{H_{CCl_4}} \left( \frac{\Delta_{CCl_4} - \Delta_f}{100} \right),$$ (2)

Here $F_{CCl_{4,parcel}}$ is the emission of $CCl_4$ across a given parcel (e.g., a 1°x1° section) of the ocean surface (mol m$^{-2}$ d$^{-1}$; negative values imply uptake), $K_w$ is the air–sea transfer velocity (m d$^{-1}$), $H_{CCl_4}$ is Henry's Law constant for $CCl_4$ (m$^3$ atm mol$^{-1}$), $p_{CCl_{4,a}}$ is its partial pressure in the atmosphere, A is the area of the parcel (m$^2$), $\Delta_{CCl_4}$ is its measured saturation anomaly (%), and $\Delta_f$ is the saturation anomaly of CFC-11. There is some uncertainty in this kind of flux estimate, mainly associated with $K_w$, which varies considerably with wind speed and sea surface roughness, but also because the CFC-11

correction is only an approximation.

The rate constant for oceanic removal of atmospheric $CCl_4$ through the parcel, $k_{CCl_{4,parcel}}$, is computed as the ratio of the flux across that surface to the total amount of $CCl_4$ in the atmosphere,

$$k_{CCl_{4,parcel}} = \frac{K_W A r_{CCl_4}}{H_{CCl_4} n_{tr}} \left( \frac{\Delta_{corrected}}{100} \right),$$ (3)

where $n_{tr}$ is the number of moles of air in the troposphere (1.46 x 10$^{20}$) and $r$ is the fraction of atmospheric $CCl_4$ that resides

in the troposphere (0.886). To account for spatial variations in surface ocean properties and surface wind speeds, we applied Eq. (3) to a monthly mean, 1°x1° gridded dataset of sea surface temperature (SST), salinity, and wind speed (Samuels and Cox 1988). Providing $CCl_4$ saturation anomalies on this same scale requires an extension of the available measurement data, since we have not made measurements in every 1°x1° grid cell. To do this, we considered correlations between the measured $CCl_4$ saturations and other properties in the 1°x1° gridded dataset (e.g., temperature, wind speed, season), but found none to

be significant in a global sense. Measured saturations do vary somewhat consistently over latitudes, so we used the mean saturation anomalies within latitudinal bands in the gridded data set to compute the global flux. This approach is not perfect—it does not capture unique coastal influences all that well, nor does it capture all areas of upwelling as a category, but it does accommodate the equatorial upwelling influences and those of some other fronts, which appear significant in the data. It also captures the temporal and spatial variability of wind speed and sea surface temperature. We ran additional sets of

computations using the global mean corrected saturation anomaly, the global median corrected saturation anomaly, and the median saturation anomaly in 10 degree latitudinal bands to test the uncertainty associated with which averaging approach



we select. Finally, we further evaluated the sensitivity of our results to these choices by alternatively considering an extreme case for the high latitude Southern Ocean, where we have little data, by selecting a saturation anomaly of -99% for that region. This last approach examines the highly unlikely possibility that circulation in the Southern Ocean could lead to extreme undersaturations and how it might influence the global flux estimates derived.

**4 Results and Discussion**

**4.2 Explaining the Observed Undersaturations**

A comparison of $CCl_4$ mole fractions measured in air from the ship's bow and from the equilibrator headspace shows that $CCl_4$ is largely undersaturated by about 5–10% in the surface ocean virtually everywhere, nearly all the time (Fig. 3, 4a). Larger undersaturations are measured in equatorial regions and, occasionally, other areas of upwelling, which is consistent

with the delivery of subsurface waters undersaturated in $CCl_4$ with respect to CFC-11. Exceptions to this general picture are periodic, small supersaturations, often, but not always, measured in coastal waters or in rough seas. These could be evidence of periodic $CCl_4$ contamination from the ship or a localized anthropogenic source of $CCl_4$, such as riverine runoff. They could also result from an inadequate correction of physical influences on the saturation anomaly. No evidence exists for production of $CCl_4$ in seawater. Three of the earlier cruises, SAGA II, RITS-89 and OAXTC-92, were conducted at a time

when contamination or analytical artifacts associated with these compounds was not uncommon and difficult to avoid, particularly in rough seas, owing to their widespread use and ubiquitous nature as solvents and in other common materials. Consequently, we have removed these positive values in our flux computations because including them would bias the results. Our goal is to determine an air–sea flux that best represents $CCl_4$ that is irreversibly removed by reactions in the ocean, which, in this instance is imperfectly represented by the corrected saturation anomaly.

These widespread undersaturations exceed those that might be expected from physical effects, such as mixing or warming of water masses. While the corrections for physical effects that we use do make the saturation anomaly more negative, the undersaturations calculated without these corrections still generally fall within 5–10%. Large departures from this range were usually associated with high mixing rates (e.g., upwelling) or rapid heating and were corrected somewhat, though not completely, with the algorithms applied here (e.g., Fig. 4a,b). Corrected saturation anomalies more negative than the -5 to -

10% range might still be due in part to differences in the physical properties of $CCl_4$ and CFC-11 that are not fully corrected in our approach (e.g., Table 2). They may also be a result of differences in the atmospheric histories of $CCl_4$ and CFC-11, where $CCl_4$ has a longer and slightly differently shaped historic accumulation in the atmosphere than CFC-11, which is reflected somewhat in its concentration-depth profiles in the ocean as additional scatter in correlation plots of these gases. Finally, trace gases in surface waters may not have always equilibrated with an atmosphere identical to that observed above

it. All in all, the corrections used here for physical effects largely reduce error and remove biases in the computation of atmospheric lifetime with respect to oceanic loss and thus lead to a better estimate (e.g., Butler et al. 1991). It's important to





note that the corrections for physical effects are most often smaller than the gross flux of $CCl_4$ into the water and therefore do not change the overall picture of "widespread undersaturation" of $CCl_4$.

Plotting the corrected saturation anomalies from our cruises shows a reasonable degree of scatter, but remarkably consistent means and medians (Fig. 5; Table 3). Larger undersaturations near the equator are likely associated with increased
upwelling, which brings up water more deficient in $CCl_4$ than CFC-11. Causes of the larger undersaturations in other areas are less clear but include, for example, a Gulf Steam ring (GasEx98), coastal waters, and larger scale ocean fronts, all of which are associated with some degree of upwelling. Yet, some areas of large, negative saturation anomalies, such as the central gyre north of Hawaii during BACPAC-99, are not so readily understood and cannot be attributed to sampling or analytical artifacts.
This relative consistency of undersaturation in surface waters during all seasons, regardless of sea surface temperature and biological regime, would suggest that $CCl_4$ is removed not in surface waters, but at depth, with the deficit advected and mixed to the surface, and ultimately to the atmosphere. First-order computations of the time required to mix waters between the surface and intermediate depths, however, suggest that, on average, the loss at depth cannot fully support the observed surface water deficits. Air–sea exchange renews gases in surface waters on the order of 20–30 days, whereas transport from
depths of hundreds of meters requires times of years to decades. Exceptions are apparent in areas of upwelling, where water from depth is advected as well as mixed toward the surface in a matter of days (e.g., Tanhua and Liu 2015). Depletion of $CCl_4$ at depth would be in agreement with other reports suggesting a loss of $CCl_4$ in low oxygen waters (e.g., Lee et al, 1999). A number of depth profiles of $CCl_4$ along with CFCs 11 and 12 on our cruises suggest a sink for $CCl_4$ in intermediate waters, typically near the oxygen minimum (Fig. 6). The relationships of the relative saturations of $CCl_4$ vs CFC-12 and
apparent oxygen utilization (AOU) throughout the water column, where AOU is the difference between the calculated atmospheric equilibrium concentration of dissolved oxygen and the measured oxygen concentration, suggest a clear, though water-mass dependent, relationship between the ratio of $CCl_4$ saturation to that of CFC-12 and AOU, particularly in waters of low oxygen concentration (high AOU; Fig. 7). This invokes the possibility of $CCl_4$ degradation by microorganisms in oxygen-deficient waters. The actual mechanisms are not discernible from these data, nor is the degree of loss in waters of
different oxygen content, though the loss would be consistent with anaerobic or heterotrophic metabolism.

So, if exchange with deeper waters where $CCl_4$ is anaerobically degraded, presumably by microorganisms (e.g., Krone et al. 1991, Lee et al 1999, Tanhua et al. 1996), cannot account for widespread undersaturations in surface waters, then there must also be some mechanism for in situ removal in oxygenated surface water. With a calculated degradation time of ~2600 y, hydrolysis does provide a removal rate that could balance air–sea exchange (e.g., Yvon-Lewis and Butler 2002, Jeffers et al.
1994), so that leaves us with invoking biological or other unknown mechanisms removing $CCl_4$ in well oxygenated surface waters. This is an area that requires further investigation, as there is no direct evidence to date for such pathways, microbial or otherwise.




### 4.2 Estimating the Air–Sea Flux

#### 4.2.1 Uncertainties

Because the observed undersaturations in surface waters appear somewhat independent of SST or surface biological activity, the major remaining variable in the distribution of the oceanic sink is the air sea exchange rate, driven largely by wind speed

and not, for example, SST, as has been observed for $CH_3CCl_3$ and $CH_3Br$ (e.g., Butler et al 1991, Yvon-Lewis et al., 1997). Our understanding of the dependence of air–sea exchange on wind speed has evolved substantially over time and studies over the past 15 years have been converging (Figure 8). Perhaps the most influential of these is by Naegler et al. (2006), which was a re-evaluation of bomb $^{14}CO_2$ uptake that has led to substantially lower estimates of air–sea exchange (Sweeney et al. 2007, Wanninkhof et al. 2009). These new studies lower the air–sea flux by 30–40% relative to Wanninkhof (1992),

which was based on earlier estimates of the ocean inventory of bomband natural $^{14}CO_2$ (e.g., Broecker et al 1985), and which we used in previous assessments of the loss of atmospheric $CCl_4$ to the ocean (Yvon-Lewis and Butler, 2002). This in turn decreases the flux and increases the calculated lifetime of the gas in the atmosphere with respect to loss to the ocean. To estimate overall flux uncertainties, we added in quadrature the standard deviation of the mean exchange rate for $CCl_4$ based on the five air–sea exchange studies we've considered (~30%; Table 2) and the standard deviation of the $CCl_4$ flux ((~14%;

Table 4) for four scenarios under one air–sea exchange relationship (Sweeney et al. 2007). Other uncertainties are either captured within these uncertainties or comparatively insignificant in the calculations.

#### 4.2.2 Air–Sea Flux and Lifetime

Propagating the rate constant for oceanic removal of atmospheric $CCl_4$ across the globe with consideration of parameters in Eq. 3 and the updated dependence of $K_W$ on wind speed, our model derives a rate constant for uptake of atmospheric $CCl_4$ by

20 the ocean that is highest where winds are strongest and turnover of surface waters fastest, but also influenced by saturation anomaly (Fig. 9). As noted in the Methods section, we used four approaches for estimating the removal of atmospheric $CCl_4$ by the ocean, but our preferred approach was using the latitudinally binned mean saturation anomalies along with the air–sea exchange relationship of Sweeney et al. (2007). We note that we obtain the same results by simply using the median of all negative corrected saturation anomalies or by averaging fluxes from all approaches.

Summing uptake across the world's oceans suggests that the oceanic sink represents a partial atmospheric lifetime of 209 (157–313) y, a number considerably longer than the 94 y of Yvon-Lewis and Butler (2002) used in the 2002–2014 WMO/UNEP Scientific Assessments of Ozone Depletion. This reduction in estimated removal rate is largely a result of the change in our understanding of the air–sea exchange coefficient. Though we have selected the latitudinally binned, global mean, corrected saturation anomaly as our best approach for interpreting the saturations, the difference among selected

approaches is relatively small (Table 4). The scenario where we tested the extreme possibility of the Southern Ocean being 99% undersaturated south of 65°S changed the flux by about 3Gg y$^{-1}$ (a ~30% increase in uptake), but, given the observed undersaturations both north and south of 65°S where we obtained measurements, that scenario is unrealistic; it was run only





to understand extreme behaviour. Barring that, and given the hemispheric asymmetries in wind speed and ocean area, the model calculates uptake by the ocean in the Southern Hemisphere that is about 1.8 times that in the northern hemisphere (Table 4).

## 5 Implications for atmospheric CCl₄

If we consider the revised oceanic sink derived here, the even weaker soil sink of 375 (288–536) y, and the partial lifetime for CCl₄ removal in the stratosphere of 44 (36–58) y, the mid-range estimate for the lifetime of CCl₄ in the atmosphere would be 33 (28–41) y, somewhat longer than the 26 (23–33) y used in the past four quadrennial assessments on ozone depletion. The ocean is responsible for removing ~16% of the CCl₄ in the atmosphere and the difference in uptake by the ocean in the two hemispheres is about 3 Gg y⁻¹ for a 90 ppt atmosphere, so can account for only about 10% of the current

interhemispheric difference of 1.2 ppt in the atmosphere (Table 4). The interhemispheric difference in CCl₄ mixing ratio is still larger than can be accounted for based on known emissions. This suggests additional emissions in the northern hemisphere, as the larger oceanic sink reported here for the southern hemisphere cannot account for it. Nevertheless, the overall budget of atmospheric CCl₄ is now much closer to being balanced, owing largely to the findings presented in this paper and to the re-evaluation of the soil sink (Happell et al 2014, Rhew and Happell 2016).

Considering this new, longer lifetime for atmospheric CCl₄ and the atmospheric trends and distributions given in Carpenter and Reimann (2014), the remaining discrepancy between potential emissions suggested by data on CCl₄ production for different uses and destruction quantities reported to the UNEP Ozone Secretariat (e.g., Montzka and Reimann et al. 2011) and emissions computed from atmospheric lifetime is now of the order of 10–20 Gg y⁻¹. While it is possible that historical natural fluxes could in part account for this additional source, no evidence exists for its presence in any significant quantity,

especially given the small mole fractions measured in the oldest firn air. Butler et al. (1999) noted that the lowest firn air values were near detection limits in their samples and suggested that CCl₄, though at times not differing from zero, could have been as high as 5–10 ppt in the atmosphere before 1900; recent, unpublished firn air data we have obtained, however, suggest that it is more likely around 3–4 ppt in the late 19th century. Consequently, most of the emission discrepancy must arise from heretofore unquantified, anthropogenic sources, predominantly in the northern hemisphere, to be consistent with

the observed rate of decline for the global mole fraction, given our understanding of the global lifetime and the mean hemispheric difference measured for atmospheric CCl₄ (e.g., Fraser et al. 2014, Liang et al 2014; Carpenter and Reimann et al., 2014). Only with an excess of northern hemispheric sources would the deficit identified in this study and its distribution be fully consistent with the observed rate of decline of CCl₄ in the atmosphere (1.2–1.4% y⁻¹), the observed interhemispheric gradient of ~1.2 ppt in recent years and an interhemispheric exchange time of the order of 1 year (Carpenter and Reimann et

al. 2014).





*Acknowledgements*. We thank the captains, officers, and crew of all of the ships involved during the entirety of this study. This research could not have been done without the support of our various institutions and the programs through which they support science, including funds at various times from NASA's Upper Atmosphere Research Program, the US Department of Energy, NOAA's Climate Program Office, the Atmospheric and Geosciences sections of the National Science Foundation, and the National Research Council of the US National Academies of Science.

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





Table 1. Cruise details including season, ship, analytical technique used to measure CCl₄, water sampling technique, and color code for Figures 1 and 5. Gas chromatography with electron capture detection (ECGC) and gas chromatography with mass spectrometry (GCMS) are the analytical techniques.

| Cruise Name | Dates | Ship | Analytical Technique | Sample Technique | Color Code |
|---|---|---|---|---|---|
| SAGA-2 | 1 May – 9 June 1987 | R/V Akademik Korolev | ECGC | Equilibrator | Dark Gray |
| RITS89 | 6 Feb – 19 April 1989 | R/V Discoverer | ECGC | Equilibrator | Light gray |
| SAGA-3 | 10 February – 12 April 1990 | R/V Akademik Korolev | ECGC | Equilibrator | Blue |
| OAXTC | 4 Aug – 21 October 1992 | R/V John V. Vickers | ECGC | Equilibrator | Red |
| BLAST 1 | 28 January – 17 February 1994 | R/V Discoverer | GCMS | Equilibrator | Navy |
| BLAST 2 | 18 October – 21 November 1994 | R/V Polarstern | ECGC,GCMS | Equilibrator | Cyan |
| BLAST 3 | 26 February – 7 April 1996 | R/V Nathaniel B. Palmer | GCMS | Equilibrator | Magenta |
| GasEx98 | 7 May – 27 July 1998 | R/V Ronald H. Brown | GCMS | Equilibrator | Black |
| RB9906 | 14 September – 23 October 1999 | R/V Ronald H. Brown | GCMS | Equilibrator | Purple |
| CLIVAR01 | 29 October – 13 December 2001 | R/V Aurora Australis | GCMS | Surface Niskin | Maroon |
| A16N | 4 June – 11 August 2003 | R/V Ronald H. Brown | GCMS | Surface Niskin | Yellow |
| PHASE | 22 May – 2 July 2004 | R/V Wecoma | GCMS | Equilibrator | Brown |
| A16S | 11 January – 26 | R/V Ronald H. Brown | GCMS | Surface | Tan |



|  |  |  |  | Niskin |  |
| --- | --- | --- | --- | --- | --- |
| GOMECC | 10 July – 4 August 2008 | R/V Ronald H. Brown | GCMS | Equilibrator | Dark Green |
| HalocAST-P | 30 May – 27 April 2010 | R/V Thomas Thompson | GCMS | Equilibrator and Surface Niskin | Orange |
| HalocAST-A | 25 October – 26 November 2010 | R/V Polarstern | GCMS | Equilibrator | Green |





Table 2. Properties of CFC-11, CFC-12 and CCl$_4$ shown as area weighted global means from our model and the Global Oceanographic Data Set Atlas, where the area-weighted global mean wind speed is 7.1 m s$^{-1}$, SST is 18.9°C, and mixed layer depth is 66.5 m. The oceanic uptake rates and partial atmospheric lifetimes were calculated using the latitudinally binned median saturation anomalies (see Table 3).

| | CFC-11 | CFC-12 | CCl4 |
|---|---|---|---|
| **Physical Properties** | | | |
| Diffusivity (D; 10$^5$ cm$^2$ s$^{-1}$) | 0.9076 | 0.9844 | 0.8465 |
| Solubility (S; m$^3$ atm mol$^{-1}$) | 0.1116 | 0.3983 | 0.0344 |
| $\Delta S/\Delta T$ (0–30°C) | 0.0044 | 0.0144 | 0.0014 |
| Trop. Mixing Ratio (ppt) (2012) | 236.0 | 524.0 | 90.0 |
| **Calculated Properties** | | | |
| **[Liss and Merlivat, 1986]** | | | |
| Gas Exchange Velocity (m d$^{-1}$) | 1.68 | 1.75 | 1.62 |
| Ocean Uptake (Gg y$^{-1}$) (2012) | 0.0 | 0.0 | 7.4 |
| Partial Lifetime (y) | NA | NA | 305 |
| **[Wanninkhof, 1992]** | | | |
| Gas Exchange Velocity (m d$^{-1}$) | 3.51 | 3.65 | 3.39 |
| Ocean Uptake (Gg y$^{-1}$) (2012) | 0.0 | 0.0 | 15.6 |
| Partial Lifetime (y) | NA | NA | 145 |
| **[Nightingale et al., 2000]** | | | |
| Gas Exchange Velocity (m d$^{-1}$) | 2.29 | 2.39 | 2.21 |
| Ocean Uptake (Gg y$^{-1}$) (2012) | 0.0 | 0.0 | 10.1 |
| Partial Lifetime (y) | NA | NA | 225 |
| **[Sweeney et al., 2007]** | | | |
| Gas Exchange Velocity (m d$^{-1}$) | 2.43 | 2.53 | 2.34 |
| Ocean Uptake (Gg y$^{-1}$) (2012) | 0.0 | 0.0 | 10.8 |
| Partial Lifetime (y) | NA | NA | 210 |
| **[Wanninkhof et al., 2009]** | | | |
| Gas Exchange Velocity (m d$^{-1}$) | 2.01 | 2.10 | 1.95 |
| Ocean Uptake (Gg y$^{-1}$) (2012) | 0.0 | 0.0 | 8.8 |
| Partial Lifetime (y) | NA | NA | 256 |



Table 3. Corrected saturation anomalies for 10 degree latitude-bins. Positive values not included (see text and Fig. 5).

| Bin | Assigned Latitude | mean | median | N | Standard Deviation | Standard Error |
|---|---|---|---|---|---|---|
| < -65 | -70 | -7.40 | -7.29 | 372 | 1.78 | 0.09 |
| -55 to -65 | -60 | -4.16 | -4.35 | 176 | 1.84 | 0.14 |
| -45 to -55 | -50 | -5.40 | -4.31 | 148 | 4.27 | 0.35 |
| -35 to -45 | -40 | -7.94 | -5.13 | 192 | 7.80 | 0.56 |
| -25 to -35 | -30 | -7.31 | -6.88 | 129 | 4.63 | 0.41 |
| -15 to -25 | -20 | -7.24 | -5.95 | 255 | 4.83 | 0.30 |
| -5 to -15 | -10 | -8.20 | -7.80 | 331 | 3.43 | 0.19 |
| -5 to +5 | 0 | -9.02 | -8.53 | 723 | 4.71 | 0.18 |
| 5 to 15 | 10 | -7.31 | -6.97 | 722 | 3.64 | 0.14 |
| 15 to 25 | 20 | -6.68 | -5.46 | 539 | 4.92 | 0.21 |
| 25 to 35 | 30 | -8.39 | -6.33 | 530 | 8.00 | 0.35 |
| 35 to 45 | 40 | -7.69 | -6.97 | 656 | 5.46 | 0.21 |
| 45 to 55 | 50 | -9.73 | -6.29 | 880 | 7.32 | 0.25 |
| > 55 | 60 | -5.56 | -5.06 | 41 | 2.67 | 0.42 |



Table 4. Calculated partial lifetimes and oceanic uptake rates from a 90 ppt atmosphere using the global median corrected saturation anomaly, global mean corrected saturation anomaly, median corrected saturation anomalies for latitude-bins, and mean corrected saturation anomalies for 10° latitude bins. The latitude bins are defined in Table 2. The partial lifetimes are independent of atmospheric amount.

| Computational Approach | Global uptake (Gg/y) | NH uptake (Gg/y) | SH uptake (Gg/y) | $\tau_{ocean}$ (y) | $\tau_{NH,ocean}$ (y) | $\tau_{SH,ocean}$ (y) |
|---|---|---|---|---|---|---|
| Global MEDIAN corrected Δ(%) | 10.8 | 3.81 | 6.98 | 210 | 338 | 151 |
| Global MEAN corrected Δ(%) | 12.6 | 3.94 | 8.62 | 180 | 290 | 130 |
| Latitudinally binned MEDIAN corrected Δ(%) | 9.0 | 3.12 | 5.83 | 252 | 365 | 192 |
| Latitudinally binned MEAN corrected Δ(%) | 10.8 | 3.81 | 6.98 | 210 | 300 | 160 |



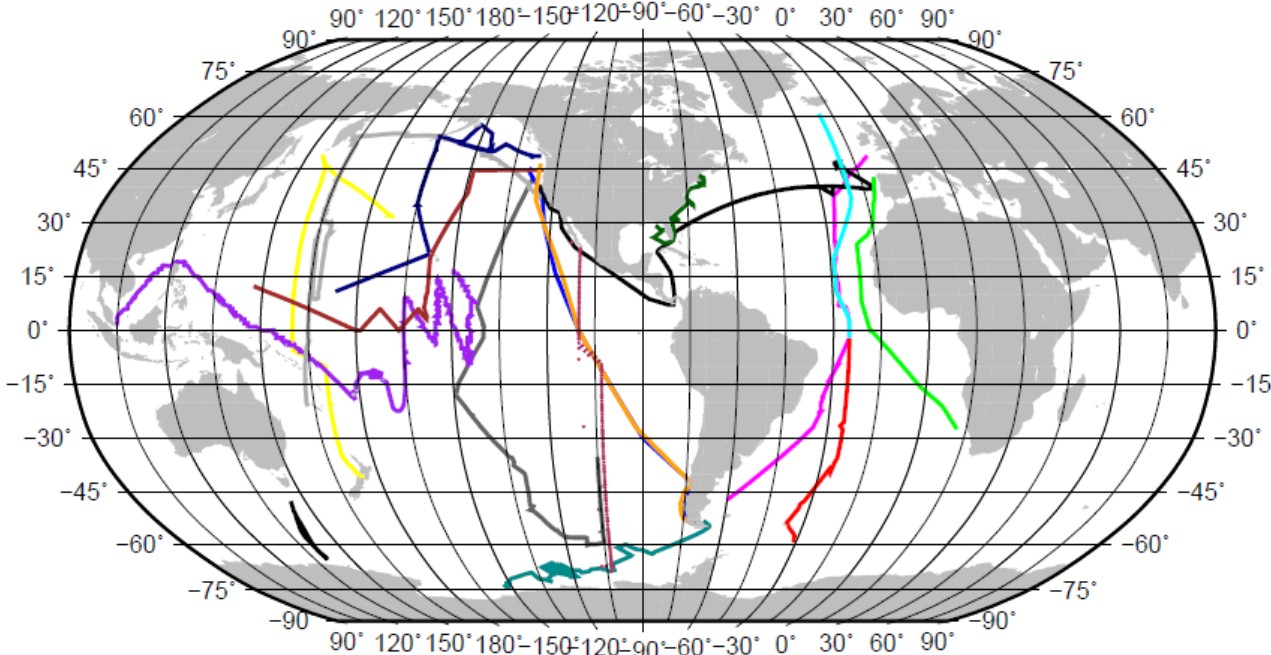

**Figure 1.** Research Cruises contributing to this study: SAGA-2, Leg I (1987, W Pacific; yellow), RITS89 (1989, E Pacific; orange), SAGA-3 (1990, Equatorial Pacific; lime) , OAXTC (1992 N. Pacific; light gray, part of WOCE P13), BLAST1 (1994, E Pacific; olive), BLAST2 (1994, Atlantic; cyan), BLAST3 (1996, Southern ocean; dark gray), GasEx98 (1998, N. Atlantic, Gulf of Mexico, NE Pacific; pink), RB9906 (1999, NE Pacific; cyan), CLIVAR01 SR3 (2001, Southern Ocean; brown) A16N (2003, N Atlantic; dark green), A16S (2005, S Atlantic; red), PHASE (2004, Central and North Pacific; purple), P18 (2008, Tropical and SE Pacific; dark green), GOMECC, Coastal NW Atlantic; olive), HalocAST-P (2010, E Pacific; blue, underlying BLAST 1, HalocAST-A (2009, E Atlantic; green).



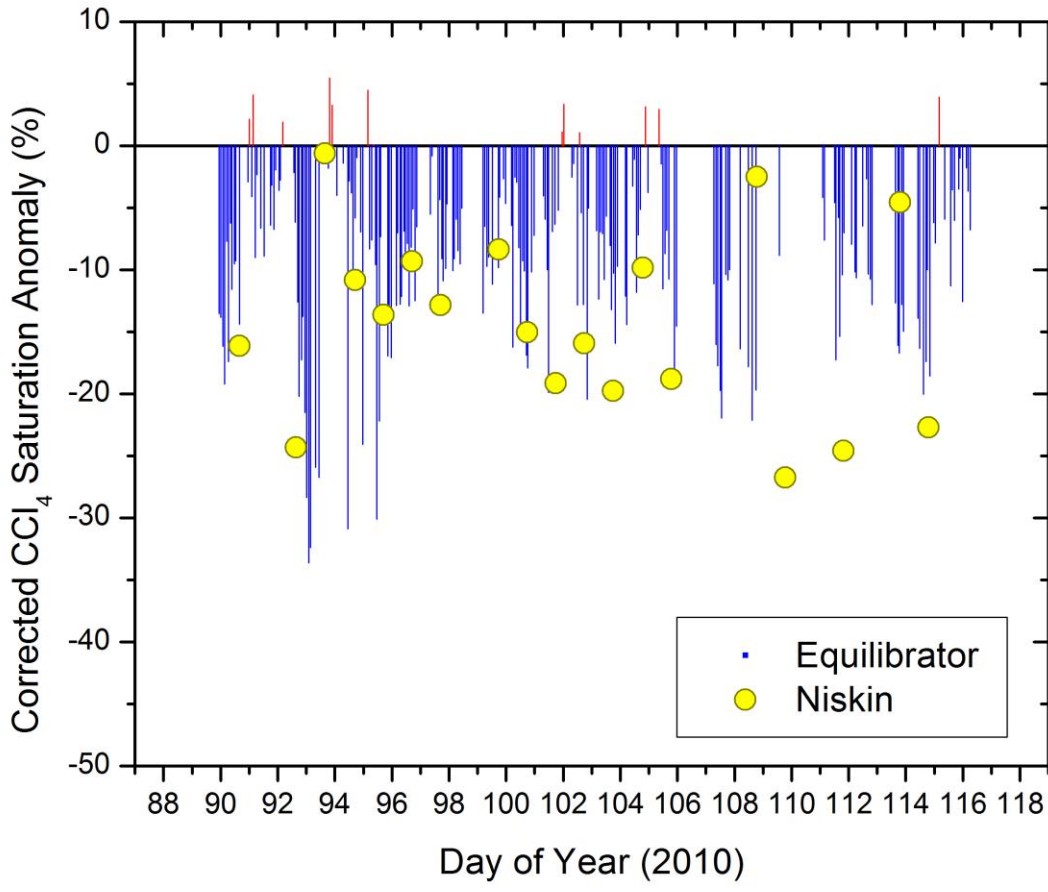

**Figure 2.** Corrected CCl₄ saturation anomalies determined from measurements of mole fractions in ambient air and air equilibrated with surface water in a Weiss-type equilibrator (blue bars) or extracted from near-surface grab samples (Niskin bottles; yellow points), showing no substantial bias in the equilibration technique for CCl₄ (Halocast-P, 2010; see Fig. 1, Table 1). Saturation anomalies in grab samples are calculated using solubility data from Bullister and Wisegarver, (1998). This good agreement suggests that the equilibrator does not bias results for CCl₄ and further suggests that the solubilities used to calculate saturation anomalies from discrete surface samples are correct.









**Figure 3.** Regionally representative examples of observed (i.e., uncorrected) $CCl_4$ undersaturations from six cruises. Corrections for physical effects can alter these anomalies either direction, but largely make them more negative by 1–3%. Negative saturations are indicated with blue lines, positive with red lines.




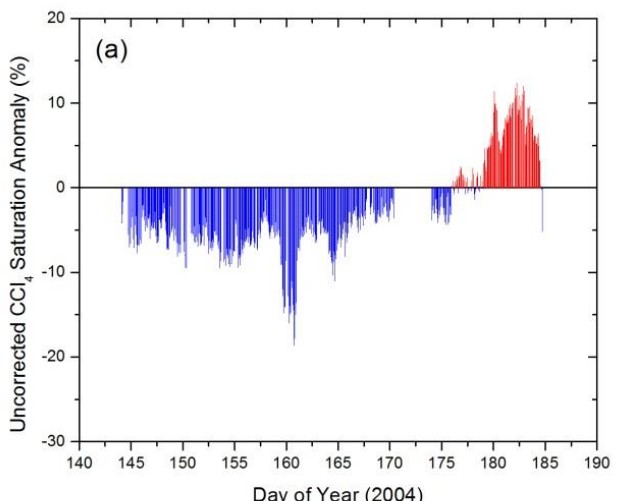
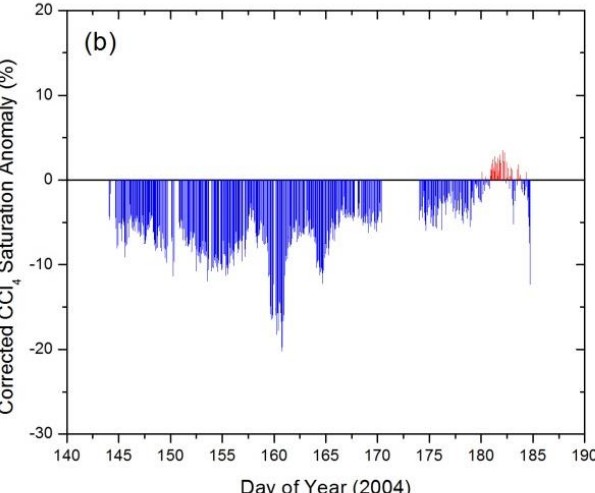

**Figure 4. Observed and corrected CCl$_4$ saturation anomalies, central and NE Pacific, show the influence of physical processes on the observed values. Corrected saturation anomalies were derived by subtracting the observed anomalies of the largely unreactive gas of similar properties, CFC-11. Negative saturation anomalies are indicated with blue lines, positive with red lines.**





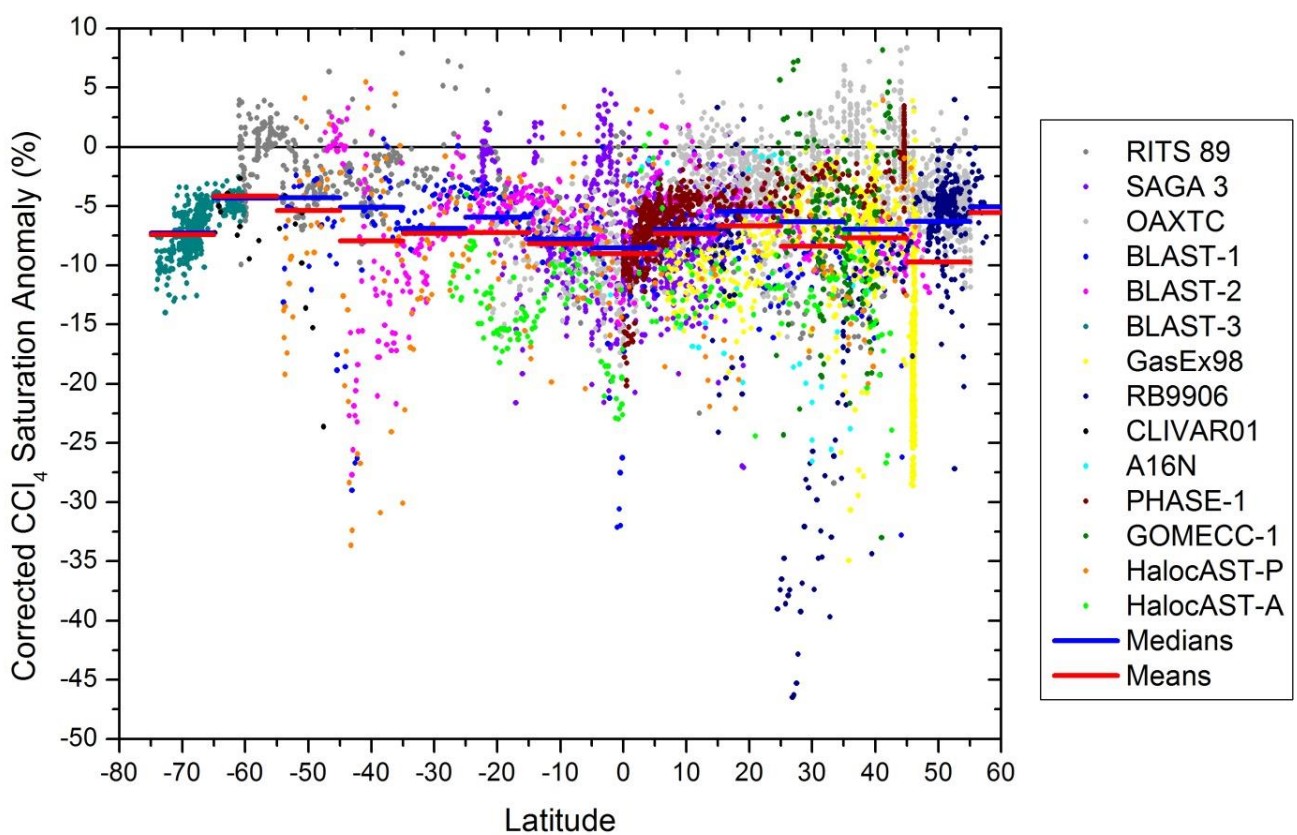

**Figure 5. Zonal distribution of the corrected saturation anomaly from all cruises. Because observed positive values are likely the result of contamination, external influences, or sampling or analytical artifacts, means and medians for each 10° latitudinal band (shown as the red and blue bars) are computed from negative saturations only (See text).**



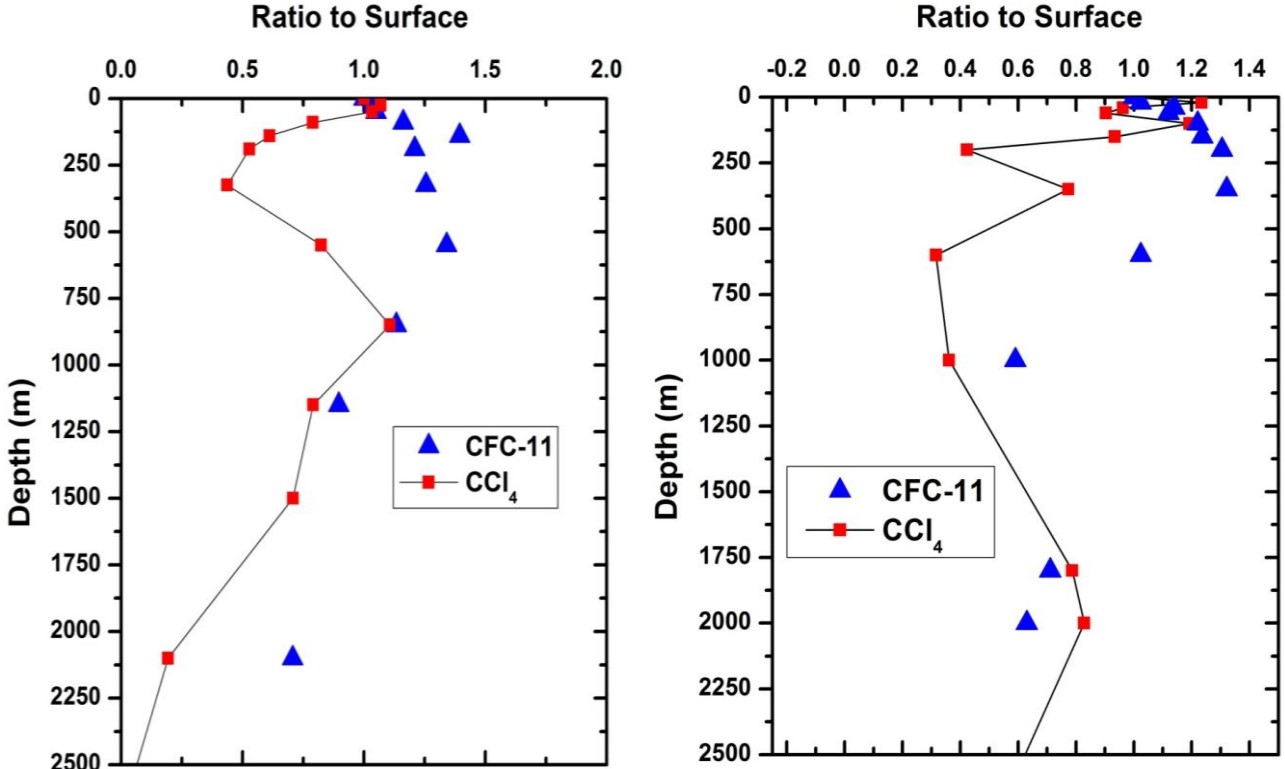

**Figure 6. Depth profiles of the concentrations of CCl₄ and CFC-11, normalized to the surface value, strongly suggest consumption of CCl₄ in intermediate waters. If there were no in situ consumption and the distributions entirely a function of mixing and transport, the curves would be similar if not identical.**



**Figure 7. Measured saturations of CCl₄ and CFC-12 in sub-surface waters during the P18 cruise in the eastern Pacific as a function of Apparent Oxygen Utilization (AOU). Each point represents a single sample extracted from a Niskin bottle and analysed by electron-capture gas chromatography. Points are colored according to the AOU in each measured sample.**





Figure 8. Various wind speed relationships for the air sea exchange velocity (k660) determined in previous studies. Curves from each of these studies were normalized to $CO_2$, but as shown here, have been adjusted to represent $CCl_4$ according to its Schmidt number, which, a function of viscosity of the medium and diffusivity of the gas, is largely a function temperature.




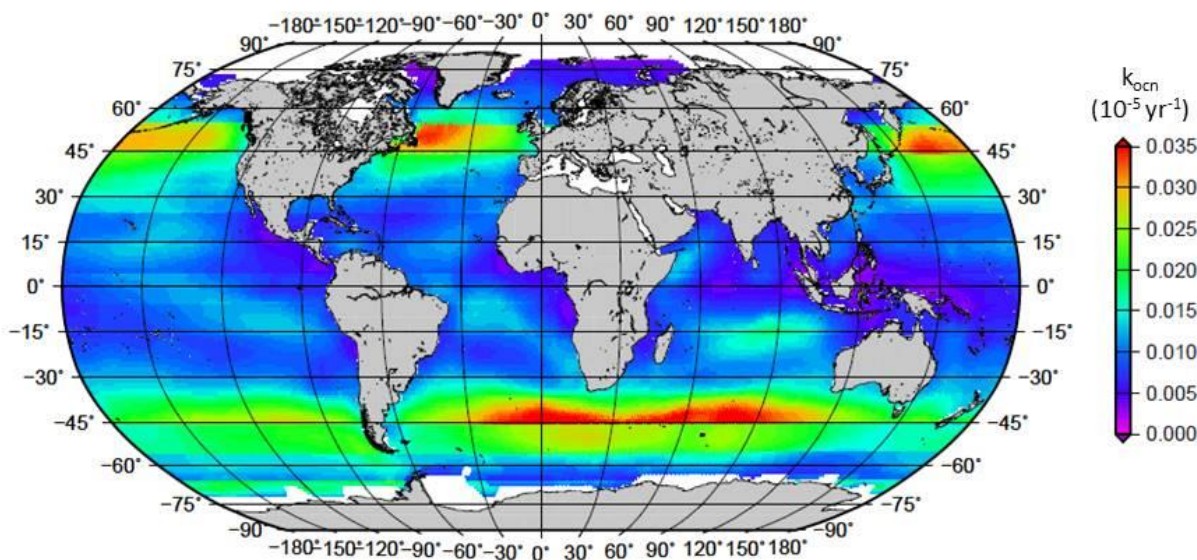

**Figure 9.** Global distribution of the oceanic uptake rate constant ($k_{ocn,i}$) based on the mean zonal saturation anomalies (Figure 5, Table 3) and calculated monthly air-sea exchange rates and other parameters given in Eq. 3 for 1° x 1° cells of the Global Oceanographic Data Set Atlas, Climatologies (Samuels and Cox 1988) . Because the saturation anomaly is largely similar among zonal regions of the ocean, variability in removal of $CCl_4$ from the atmosphere is controlled primarily by wind speed, which is highest in higher latitudes of both hemispheres.