# Peer review of "A comprehensive estimate for loss of atmospheric carbon tetrachloride (CCl4) to the ocean"

_Atmospheric Chemistry and Physics, 2016_

## Referee Comment (RC1) · Anonymous Referee #1 · 18 Apr 2016

General comments This is a high quality, well written manuscript on the important topic of how much atmospheric CCl4 is taken up by the oceans. CCl4 is an important ozone depleting compound that currently has an unbalanced atmospheric budget and any refinement of source of sink strengths will help in balancing the budget. This paper uses more data and a better estimate of air-sea exchange rates to refine the estimate the ocean sink for CCl4 and should be published. Specific comments 1) Page 2, lines 6 and 7. While the data supports the statement that the soil sink for CCl4 is less than the ocean sink the data does not support the statement that the soil sink strength is less certain. This paper states that the ocean partial lifetime is 209 (157–313) y or in percent terms 209 (-23.4%, +49.8%), while the most recent estimate of the soil sink

[Figure]

**[ACPD]{.underline}**

Interactive
comment

strength is 375 (288–536) y or 375 (-23.2% +42.9%). I suggest rewording this sentence to remove any statement on relative uncertainties between these two sinks, because they are about the same. 2) Page 2, Lines 24-25. I suggest changing "we resorted to sampling daily" to "we sampled daily" 3) Page 5, line 19, I suggest changing,"imperfectly represented" to "estimated". 4) Page 6, lines 30-32. While there is no direct evidence for the mechanism of CCl4 removal in well oxygenated surface water, there is evidence for microbial removal in well oxygenated soils. I suggest working in a statement to this would be beneficial to this paper. Use the following citation: Mendoza, Y., K.D. Goodwin, J.D. Happell, Microbial removal of atmospheric carbon tetrachloride in bulk aerobic soils, Appl. Environ. Microbiol., 77, 5835-5841, 2011.

Technical corrections 1) Page 7 line 10, "bomband" should be "bomb and". 2) Page 7 line 14, "((" should be "(". 3) Page 11 line 25, "SF 6" should be "SF6" 4) I do not think that Table 1 is needed. Most of the information given in Table 1 is in the caption of figure 4. A small expansion of the caption of Figure 4 could be made to include all information given in Table 1.

---

## Referee Comment (RC2) · Anonymous Referee #2 · 21 May 2016

This is a very welcome and important study reporting a revised partial ocean lifetime for CCl4. These authors are the only ones in the world with a sufficiently large and high quality data set of ocean CCl4 (and CFC11), alongside the relevant expertise, for such a global study, and their new data and analyses are significant and timely.

The new results are carefully reported and analysed, and the manuscript is generally extremely well written and clear. I have no major criticisms, but find that in several places there could be more detailed explanations of uncertainties and computations, as detailed below. I also recommend some discussion of the study of Huhn et al. (Deep Sea Research, 2001) which deduced that oceanic CCl4 depletion was faster at warmer temperatures (contrary to the present study where there was no dependence

on temperature), and deduced a depletion rate of approximately 22% per year for temperatures above13oC.

Pg 3 – Computations. The temperature dependence of the solubility of CFC11 is about a factor of 4 different from CCl4 from 0-30 oC (Table 2). There is an even greater absolute difference in solubility. Surely the temperature dependence of the diffusivities is also important? –ideally being the same for 2 gases. How much do all these differences compromise the use of CFC11 as a way to determine irreversible loss of CCl4 by difference?. It would be good to see some quantitative analysis of this. This would help the reader to quantitatively understand the statement later on (pg 4, line 5) that "because of the differences in physical properties of various gases, in situ consumption is more probable if the corrected saturation anomaly is less than -2%."

Pg 5, Ln 21: "While the corrections for physical effects that we use do make the saturation anomaly more negative," I'm intrigued by this. Hypothetically, if the data coverage was 100%, wouldn't the physical effects cancel because there would be as many instances of water masses radiatively cooling as radiatively warming? This is probably a very naïve assumption (I'm not an oceanographer). But it would be good to see an explanation as to why the physical effects act to overall make the saturation anomaly more negative.

Pg 6. Lns 12-14 "First-order computations of the time required to mix waters between the surface and intermediate depths, however, suggest that, on average, the loss at depth cannot fully support the observed surface water deficits.". It's not exactly clear how the estimates of mixing time scales from surface and deeper waters support this hypothesis.

Page 7 ln 20 onwards. The newly calculated ocean sink has changed by more than a factor of 2 compared to the previous assessment of 94 years. The sink is directly proportional to Kw, which is 30-40% smaller in this study than that of Wanninkhof 1992 used previously. The new analysis apparently has a very similar partial pressure

difference (air/sea) of CCl4 used previously to estimate the 94 year lifetime. So – why has the sink changed by a factor of 2 rather than only 30-40%? Even at higher wind speed conditions, which might dominate the overall sink, the differences in Kw are nowhere near a factor of 2. I think this deserves some detail in the explanation.

Minor corrections: Page 7 ln 10 – needs space between "bomband" Eqn (3) – Dcorrected is introduced, which should be defined (DCCl4$-\Delta$ðİŚŞ) Fig 8 caption – needs fixing (typos). Also, please describe the separate studies used in the legend within the caption.
* * *

---

## Author Comment (AC5) · 3 Aug 2016

Figure 8. Various wind speed relationships for the air sea exchange velocity (Kw,660) determined in previous studies. Curves from each of these studies were normalized to CO2, but as shown here, have been adjusted to represent CCl4 according to its Schmidt number, which, a function of viscosity of the medium and diffusivity of the gas, is largely a function of sea surface temperature.

[Figure]

[Figure]

**Fig. 1.**

---

## Author Response (AR1)

Response to Referee #1 Comments, "A comprehensive estimate for loss of atmospheric carbon tetrachloride ( $CCl_4$ ) to the ocean", by J.H. Butler et al.

- 1. Page 2, lines 6 and 7. The referee is correct that this is a mischaracterization of the relative uncertainty of the soil sink. We have removed the reference to the soil sink being less certain.
- 2. Page 2, lines 24-25, Done; changed to "we sampled daily"
- 3. Page 5, line 19, Done; changed to "estimated"
- 4. Page 6, lines 30-32, Done as requested. The statement reads as, "This is an area that requires further investigation, as there is no direct evidence to date for such pathways, microbial or otherwise, in the ocean, although there is evidence for microbial removal of CCl4 in well oxygenated soils (e.g., Mendoza et al., 2011)"
- 5. Technical corrections
  - a. Page 7, line 10, Text now reads "bomb and"
  - b. Page 7, line 14, Text now reads "(~14%)"
  - c. Page 11, line 25, Text now reads "SF6"
- 15d.Table 1. (The referee refers to "Figure 4" in making this comment, but we believe the reference was to
"Figure 1", which is relevant to his or her point.) After consideration, we have decided to keep Table 1 in
the text and leave the Figure 1 caption as is. The information in the caption refers only to what's in the
figure; information in the table addresses additional items such as ship names and sampling and analytical
- 20 Response to Referee #2 Comments, "A comprehensive estimate for loss of atmospheric carbon tetrachloride (CCl4) to the ocean", by J.H. Butler et al.

techniques, which are relevant to the data collected and can be linked to other studies.

Temperature trends. The referee is correct that Huhn et al (2001) observed a temperature trend in computed CCl4 loss at depth. However, Huhn et al. (2001) also noted that losses were not observed in waters of > 200 umol/kg. Our concern here is that there is no clear temperature dependence in surface waters, which implies a different process for removal. Surface waters are well ventilated so, a priori, we should expect no degradation. The fact that we do see a deficit suggests yet another process at work. We don't think that the temperature trends noted by Huhn et al. (2001) are relevant to surface concentrations, especially, as we note below, since we cannot explain the observations by transport from depth. That being said, we have added a notation of the loss rates observed by Huhn in the manuscript near the end of Section 4.2 to support our point that degradation at depth cannot explain the observed undersaturations.

1

2. Corrections for physical effects.

5

10

25

- a. **Page 3 Diffusivity temperature dependence.** The referee is correct that the change in diffusivity is about the same for the two gases (~2.7%/degree). We have added that property to Table 2.
- b. Page 3, Computations. The referee makes a good point about the temperature dependence of the solubility of the gas, which highlights the fact that we weren't presenting it correctly in the table. Because the saturation anomaly itself is expressed as percent, we should have in the table a percent increase of solubility with temperature, not absolute units of solubility. The percent change in solubility of the two gases is about the same (3.9%/degree for CFC-11 and 4.1%/degree for CCl4), which means that one should expect roughly the same influence of a change in temperature on the saturation anomaly of each gas. Thus, if one sees an increase in the CFC-11 saturation anomaly owing, for example, to radiative heating, CCl4 should respond similarly.
- c. Quantitative Analysis to support statement on page 4, line 5, and corrections for physical effects page 5, line 21. Providing a quantitative description is difficult to do without making significant assumptions about depth, extent, and timing of air injection, sea surface roughness, etc. Dissolution of bubbles favors invasion of a gas over evasion and would thus tend to sustain a positive anomaly. These effects are explained in considerable detail in Kester (1975). But, in the end, gases with similar diffusivities and changes in solubility with temperature should have similar saturation anomalies at the ocean surface if there are no other forces (e.g., production or degradation) at play. On cruises where we have measured CFC-11 and CFC-12, both long-lived in the atmosphere and conservative in surface waters, we generally see similar supersaturations and undersaturations (Lobert et al., 1995, Butler et al. 1988). The differences can be up to 2%, which we consider to be due to differences in physical properties of the two gases.

The text on Page 4 now reads as,

"Calculated this way, a corrected saturation anomaly that is negative indicates that the gas is probably being consumed in the water, regardless of its non-corrected anomaly. In some of our studies we noted that saturations of CFC 11 and CFC 12, which also have similar physical properties (Table 2), could differ by as much as ~2% (Butler et al. 1988, Lobert et al. 1995). As a result, we consider in situ consumption of CCl4 significant if the corrected saturation anomaly is more negative than -2%.

The text on Page 5 now reads as,

"The correction for physical effects that we use here, i.e., subtracting the CFC-11 saturation anomaly, which more often than not is positive, makes the  $CCl_4$  saturation anomaly more negative. Although one might expect the effects of warming and cooling to balance out on a global basis, effects such as dissolution of bubbles and mixing of

15

5

20

25

waters tend to elevate surface saturation anomalies of all gases (e.g., Kester 1975, Bowyer and Woolf 2004). Nevertheless, the undersaturations calculated without these corrections still generally fall within 5–10%."

3. **Mixing Time scales, page 6, lines 12-14.** The average eddy diffusion coefficient through the thermocline is about 1 cm2 s-1 (8.64 m2 d-1). At this rate, vertical transport through the ocean thermocline, which has a scale length of hundreds of meters to reach the nadir in CCl4 concentrations, can take tens of days to move one meter. That's not nearly fast enough to sustain a 5-10% deficit of a gas in surface waters that are replenished from the atmosphere every 20-30 days. It is also consistent with the fact that Huhn et al (2001) and others suggest degradation rates in low oxygen waters of around 2-3% per year. To make this more clear, we have revised the text as follows:

"Air–sea exchange renews gases in surface waters on the order of 20–30 days, whereas, with an eddy diffusivity of ~1 cm2s-1 through the thermocline (Quay and Stuiver, 1980; Li et al, 1984), transport to depths of hundreds of meters from the ocean surface requires times of years to decades. Exceptions are apparent in areas of upwelling, where water from depth can be advected as well as mixed toward the surface in a matter of days (e.g., Tanhua and Liu 2015). Depletion of CCl4 at depth would be in agreement with other reports suggesting a loss of CCl4 in low oxygen waters (e.g., Lee et al, 1999), although the rate of a few percent depletion per year (e.g., Huhn et al, 2001, Min et al, 2010) at depth is still not sufficient to sustain the observed undersaturations at the surface."

- 4. Newly calculated ocean sink has dropped substantially, page 7, line 20 ff. We agree, this is a big change and it's not entirely due to the air-sea exchange coefficient. We thank the author for raising this issue, as our explanation in the text was lacking. In addressing this we've also looked carefully at other air-sea exchange studies since Sweeney et al. (2007) and Wanninkhof et al. (2009). These recent studies seem to be converging around a number very near that of Nightingale et al (2000) and Sweeney et al. (2007), which was used in the original version of this manuscript. As Wanninkhof (2014) summarizes this progress and offers a complete re-evaluation of uncertainties in estimating global fluxes, we have now chosen to use the Wanninkhof (2014) estimate, normalized to the wind speeds in our model. This increases the flux of CCl4 to the ocean by about 15% over that of our original number based on Sweeney (2007), and reduces the uncertainty in estimates of the air-sea exchange coefficient to ±20%, down from the ±30% we had calculated in the previous version and the ±32% given in Sweeney (2007). The total atmospheric lifetime drops from our estimate of 33 y in the ACPD version of this paper down to 32 y. The text now reads as follows:

"This updated estimate is based on four times as many observations as used in Yvon-Lewis and Butler (2002), which account for all seasons and cover almost all major ocean basins. The average saturation anomaly used in this study is 10-20% less than the average used in Yvon-Lewis and Butler [2002]. Binning the surface data in our preferred approach (rather than applying a global mean anomaly as done before) to reflect better the actual distribution over the oceans accounts for another 10-20% decrease (Table 3). The model used by Yvon-Lewis and Butler [2002] was based the  $2^{\circ}x2^{\circ}$  COADS data set for sea surface temperatures and wind speeds and our new estimate is based on a different, newer data set with  $1^{\circ}x1^{\circ}$  bins. The mean or median wind speed for the  $1^{\circ}x1^{\circ}$  data set is ~5% lower and winds were distributed differently than in the COADS data set. The most influential change, however, is the use of an updated air-sea exchange coefficient, based on a revised inventory of bomb-14CO2 (Naegler et al 2006, Wanninkhof, 2014). Yvon-Lewis and Butler (2002) used the Wanninkhof (1992) relationship, which was normalized to an earlier assessment of bomb- $^{14}$ CO2. We evaluated the impact of this change on CCl4 flux over the ocean and determined that it alone accounts for a 24% lower flux with Wanninkhof (2014: Table 2) than with Wanninkhof (1992). Additional reductions came from use of a simpler computational approach that differs from that of Yvon-Lewis and Butler (2002), which was designed for gases where in situ loss rates are known and which required estimates of mixed layer depth and loss during downward mixing through the ocean thermocline. The newly revised estimate for  $CCl_4$  uptake provided here is based simply on the air-sea difference in partial pressure and the kinetics of air-sea exchange. It is more robust for this gas, for which there is little understanding of the loss mechanisms, and suggests that the ocean sink is responsible for about 18% (vs. 32% previously) of the  $CCl_4$  removed from the atmosphere."

Because the ACPD version has been used already in a SPARC report (*SPARC Report on the Mystery of Carbon Tetrachloride. Q. Liang, P.A. Newman, S. Reimann (Eds.), SPARC Report No. 7, WCRP-13/2016.*), we also added text to explain the difference between our submitted version and this one and give the reasoning for overriding our earlier decision to use the Sweeney et al relationship. That text (p.8) reads as follows:

"(Note: Our original version of this paper (doi:10.5194/acp-2016-311) preferred the Sweeney et al. (2007) relationship for computing air-sea fluxes and subsequent lifetimes. We had selected that parameterization because it was formulated similarly to Wanninkhof (1992), which had been used in the earlier calculations of the ocean sink of  $CCl_4$  (Yvon-Lewis and Butler 2002), accounted for the change in ocean bomb-14C inventory, and was centered among the distribution of wind speed relationships considered (e.g., Figure 8). We have since updated that and prefer the Wanninkhof (2014) polynomial approach, which includes a rigorous evaluation of the biases and uncertainties in estimating air-sea exchange, and takes additional studies into account.)"

30

5

10

15

20

25

- 5. Minor Corrections:
  - a. Page 7, line 10, space between "bomb" and "and". Done
  - b. D(corrected) needs definition. Done. Text after Eq. 3 now reads as "where  $n_{tr}$  is the number of moles of air in the troposphere (1.46 x 1020), r is the fraction of atmospheric CCl4 that resides in the troposphere (0.886), and  $\Delta_{corrected}$  is the difference between  $\Delta_{CCl4}$  and  $\Delta_{f}$ ."
  - c. Figure 8 caption typos need fixing. Done
  - d. Describe the separate studies used in the legend within the caption. Done by revising the legend to properly reference each study and with additional studies included in the plot.

**10 Relevant Changes in the Manuscript**

- 1. Lifetimes changed in the abstract and text, owing to use of an updated air-sea exchange relationship. (p. 6, 14, )
- 2. Explanation of what constitutes physical processes (p. 8, 9, 10-11)
- 3. Explanation of rates of subsurface mixing (p. 11)
- 4. Revised flux uncertainty (p.12)
- 15 5. Explanation of difference between air sea exchange coefficients in this version vs. the originally submitted version in ACPD. We felt the need to include this, as the ACPD version has already been used and cited in a SPARC report on the mystery of atmospheric CCl4 (p. 13)
  - 6. Explanation of large difference in this estimate vs the estimate given in Yvon-Lewis and Butler (2002), (p. 13-14)

20

**A comprehensive estimate for loss of atmospheric carbon tetrachloride (CCl4) to the ocean**

J. H. Butler1, S. A. Yvon-Lewis2,7, J. M. Lobert3,7, D. B. King4,7, S. A. Montzka1, J. L. Bullister5, V. Koropalov6, J. W. Elkins1, B. D. Hall1, L. Hu1,2 and Y. Liu2,8

[revised manuscript text omitted]

Table 4. Oceanic uptake rates and partial atmospheric lifetimes with respect to ocean uptake from a 90 ppt atmosphere using the Wanninkhof (2014) air-sea exchange parameterization applied to the global median corrected saturation anomaly, global mean corrected saturation anomaly, median corrected saturation anomalies for 10° latitudinal bins, and mean corrected saturation anomalies for 10° latitudinal bins. The latitudinal bins are defined in Table 3. The variability in these results demonstrates the uncertainty associated with computational approach. The partial lifetimes are independent of atmospheric amount.

5

|                              | Global | NH     | SH     |                           |                             |                              |
|------------------------------|---------------|---------------|---------------|---------------------------|-----------------------------|------------------------------|
| Computational         | uptake | uptake | uptake |  tocean | τNH,ocean |  tSH,ocean |
| Approach                     | (Gg/y) | (Gg/y) | (Gg/y) | (y)                | (y)                  | (y)                   |
| Global Median                | 12.3          | 3.9           | 84            | 184                       | 581                         | 270                          |
| corrected Δ(%)        | 12.5   |        | 0      | 104                       | 501                  | 270                   |
| Global Mean           | 14 4          | 46            | 9.8           | 157                       | 492                         | 231                          |
| corrected Δ(%)        | 14.4   |        |        | 107                |                      | 201                   |
| Latitudinally binned,        | 10.3          | 3.6           | 67            | 220                       | 629                         | 338                          |
| Median corrected Δ(%) | 10.5   | 5.0    | 0.7    | 220                       | 027                         |                       |
| Latitudinally binned         | 12.4          | 44            | 8.0           | 183                       | 514                         | 283                          |
| Mean corrected Δ(%)   | 14.7          |        | 0.0           | 105                       |  717                 | 203                          |

10

Table 4. Calculated partial lifetimes and oceanic uptake rates from a 90 ppt atmosphere using the global median corrected saturation anomaly, global mean corrected saturation anomaly, median corrected saturation anomalies for latitude bins, and mean corrected saturation anomalies for 10° latitude bins. The latitude bins are defined in Table 2. The partial lifetimes are independent of atmospheric amount.

|                               | <del>Global</del> | NH                | <del>SH</del>     |                        |                       |                             |
|-------------------------------|-------------------|-------------------|-------------------|------------------------|-----------------------|-----------------------------|
| Computational          | <del>uptake</del> | <del>uptake</del> | <del>uptake</del> |                        | Ŧ NH,ocean | ∓SH,ocean |
| Approach                      | <del>(Gg/y)</del> | <del>(Gg/y)</del> | <del>(Gg/y)</del> | τ ocean (y) | <del>(y)</del>        | <del>(y)</del>              |
| GlobalMEDIANcorrected A(%)    | <del>10.8</del>   | <del>3.81</del>   | <del>6.98</del>   | <del>210</del>         | <del>338</del>        | <del>151</del>              |
| Global MEAN
corrected Δ(%) | <del>12.6</del>   | <del>3.94</del>   | <del>8.62</del>   | <del>180</del>         | <del>290</del>        | <del>130</del>              |
| Latitudinally binned          | <del>9.0</del>    | <del>3.12</del>   | <del>5.83</del>   | <del>252</del>         | <del>365</del>        | <del>192</del>              |

| MEDIAN corrected     |                 |                 |                 |                |                |                |  |
|----------------------|-----------------|-----------------|-----------------|----------------|----------------|----------------|--|
| ∆(%)          |                 |                 |                 |                |                |                |  |
| Latitudinally binned |                 |                 |                 |                |                |                |  |
| MEAN corrected       | <del>10.8</del> | <del>3.81</del> | <del>6.98</del> | <del>210</del> | <del>300</del> | <del>160</del> |  |
| <del>∆(%)</del>      |                 |                 |                 |                |                |                |  |